

# Particle settling and convective mixing in the Saharan Air Layer as seen from an integrated model, lidar, and in-situ perspective

Josef Gasteiger[1,2], Silke Groß[3], Bernadett Weinzierl[2], Daniel Sauer[3], and Volker Freudenthaler[1]

[1]Meteorologisches Institut, Ludwig-Maximilians-Universität, München, Germany.
[2]Faculty of Physics, University of Vienna, Vienna, Austria.
[3]Institut für Physik der Atmosphäre, Deutsches Zentrum für Luft- und Raumfahrt, Oberpfaffenhofen, Germany.

*Correspondence to:* Josef Gasteiger (josef.gasteiger@univie.ac.at)

**Abstract.**

Long-range transport of aerosol in the Saharan Air Layer (SAL) across the Atlantic plays an important role for weather, climate, and ocean fertilization. However, processes occurring within the SAL and their effects on aerosol properties are still unclear. In this work we study particle settling and convective mixing within the SAL based on measured and modeled vertical aerosol profiles in the upper 1 km of the transported SAL. We use ground-based POLIS lidar measurements and airborne particle counter measurements over the Western Atlantic, as well as space-based CALIOP lidar measurements from Africa to the Western Atlantic. In our model we take account of the optical properties and the Stokes gravitational settling of irregularly-shaped Saharan dust particles.

We test two hypotheses about the occurrence of convective mixing within the SAL over the Atlantic to explain the aerosol properties observed by the lidars and the particle counter. Our first hypothesis (H1) assumes that no mixing occurs in the SAL leading to an altitude separation of super-micron dust particles as a result of settling. The second hypothesis (H2) assumes that convective mixing occurs in the SAL during the day allowing large super-micron dust particles to stay airborne longer than without convective mixing.

In general, a decrease of the particle linear depolarization ratio towards the SAL top is found in the measured lidar data but the decrease is much weaker than modeled in case of H1. The in-situ data on particle number concentrations show a presence of large particles near the SAL top that is inconsistent with H1. Furthermore, the analysis of the CALIOP measurements reveals that the average vertical profile of the linear depolarization ratio of the aerosols in the upper 1 km of the SAL does not change along its transport path over the Atlantic. These findings indicate H2 to be much more likely than H1, giving evidence that convective mixing occurs within the SAL over the Atlantic with significant consequences for the evolution of the size distribution of the super-micron dust particles during transport.

## 1 Introduction

The Saharan Air Layer (SAL) carries large amounts of Saharan aerosol particles towards the Western Atlantic Ocean and the Americas, in particular during summer time (Prospero and Carlson, 1972; Carlson and Prospero, 1972; Schütz, 1980). The SAL over the African continent often is a well-mixed convective layer from the hot surface to about 4-6 km above sea level




(Ben-Ami et al., 2009; Knippertz et al., 2009; Cuesta et al., 2009; Esselborn et al., 2009). As soon as the SAL reaches the Atlantic, it is lifted over a comparatively cold marine boundary layer. As a consequence, the radiative heating at the bottom of the SAL, which is a strong driving force for convection, vanishes. However, weaker convection within the SAL over the Atlantic might be possible due to other radiative or dynamic effects. Knowledge about those processes is quite limited but they can be important for the evolution of the particle size distribution during transport. Changes in size distribution can have significant effects for radiative properties and deposition of Saharan aerosols (e.g., Otto et al., 2009; Mahowald et al., 2014).

Size distribution measurements performed at Izaña (Canary Islands) and Puerto Rico (Caribbean) by Maring et al. (2003) revealed that Saharan dust particles with $r > 3.6$ µm are preferentially removed during the transport over the Atlantic. Maring et al. (2003) cannot explain their measurements with Stokes gravitational settling alone, but they have to reduce the Stokes settling velocity by 0.0033 m s$^{-1}$ to match the measurements. This could be an indication for vertical mixing of air during the transport.

Lidar remote sensing is a powerful tool to localize and characterize aerosols, including their size distributions. The particle linear depolarization ratio $\delta_l$ (Sassen, 1991), measured by advanced lidar systems, is a particularly useful parameter to characterize Saharan aerosols. For example, Liu et al. (2008) characterize in a case study a dust outbreak that was transported from the Sahara over the Atlantic. They use measurements of the CALIOP lidar (Winker et al., 2009), which is operated on-board the CALIPSO satellite and measures $\delta_l$ at a wavelength of 532 nm. Liu et al. (2013) investigate Asian dust and its transport over the Pacific using data from the same instrument. The network EARLINET (Pappalardo et al., 2014) provides a comprehensive data set on ground-based lidar measurements throughout Europe, which is useful to study Saharan aerosols transported to Europe (see e.g., Mattis et al., 2002; Papayannis et al., 2008; Wiegner et al., 2011). During field campaigns like PRIDE (Reid et al., 2003), SAMUM (Heintzenberg, 2009; Ansmann et al., 2011), and SALTRACE (Weinzierl et al., 2016), Saharan aerosol has been measured using a wide set of techniques, including lidar, photometer, and airborne in-situ instrumentation. The combination of different measurement techniques enables one to better constrain the properties of the rather complex Saharan aerosol. Polarization-sensitive (near-)backscattering by dusty aerosols is studied also in laboratories (e.g., Sakai et al., 2010; Järvinen et al., 2016).

Yang et al. (2013) investigate Saharan aerosols on their way over the Atlantic based on $\delta_l$ data from CALIOP. They use $\delta_l$ from volumes that the CALIPSO operational algorithm classified as dust-laden and average $\delta_l$, as function of height above sea level, over the summer season 2007. The averaged $\delta_l$ profiles show an increasing height dependence with increasing distance from Africa. In the Western Atlantic they find the largest $\delta_l$ values at altitudes of about 4-5 km and a decrease of $\delta_l$ with decreasing altitude. Yang et al. (2013) explain the averaged CALIOP $\delta_l$ profiles with shape-induced gravitational sorting using a simple model which assumes that particles with nearly spherical shape fall faster and have smaller $\delta_l$ than particles with stronger deviation from spherical shape.

In our study we investigate the Saharan aerosol transport over the Atlantic by combining advanced modeling efforts with data obtained from ground-based lidar, from airborne particle counters, and from the CALIOP lidar. We show theoretical profiles for Saharan aerosols considering gravitational settling as function of particle size and shape, and two hypotheses about the occurrence of convective mixing within the SAL. The lidar-relevant optical properties are simulated based on the particle





microphysics explicitly using an optical model. We compare our modeled profiles with the measured data that we evaluate as function of distance from the SAL top. The ground-based and airborne measurements used in our study were performed during the SALTRACE field campaign in summer 2013 in the vicinity of Barbados (Caribbean). From CALIOP, we use night-time profile data covering 15 summer months and the Saharan aerosol transport region from Africa to the Caribbean.

After describing our modeling approach (Sect. 2) we investigate modeled profiles after five days of transport without convective mixing (Sect. 3.1), which is a typical transport time of Saharan aerosol to the Caribbean. The sensitivity of $\delta_l$ profiles to particle shape (Sect. 3.2) and to the shape dependence of the settling velocity (Sect. 3.3) is investigated subsequently. In Sect. 3.4 we model the effect of day-time convective mixing occurring in the SAL during transport. Subsequently, the modeled profiles are compared in a case study to lidar and in-situ profiles measured in Barbados during SALTRACE (Sect. 4) to test

our two hypotheses about convection in the SAL over the Atlantic. In Sect. 5 we continue testing these hypotheses by using averaged SALTRACE lidar profiles and averaged CALIOP profiles.

## 2   Model description

Our model describes Saharan aerosols in the SAL. We consider six irregular dust particle shapes, as introduced by Gasteiger et al. (2011), with shapes A-C being deformed prolate spheroids with varying aspect ratio, shape D an aggregate, and shapes

E-F edged particles with varying aspect ratio (these shapes are depicted further down in Fig. 5). We assume the particles to be randomly oriented. We furthermore assume that the SAL initially is well-mixed and that gravitational settling of the aerosol particles is the only process when convective mixing is stopped. We also consider the case of a diurnal cycle of the convective mixing activity.

### 2.1   Stokes settling

The settling velocity $v$ of a particle relative to the surrounding air is determined mainly by the balance between gravitation force $F_g$ and drag force $F_d$. The gravitation force is given by

$$F_g = \frac{4}{3}\pi r_v{}^3 \rho g \tag{1}$$

with the volume-equivalent radius $r_v$ of the particle, the gravitational acceleration $g = 9.81$ m s$^{-2}$, and the particle density $\rho$ that we assume to be $2.6 \cdot 10^3$ kg m$^{-3}$ for desert dust particles. The drag force of an aerosol particle in the size range from $r \approx$

$0.5$ µm to $r \approx 10$ µm (being in the Stokes' drag regime) can be approximated by

$$F_d = 6\pi \eta r_c v \tag{2}$$

with the dynamic viscosity of air $\eta = 17$ µPas (approx. value for $0°$C temperature and tropospheric pressures) and the cross-section-equivalent radius $r_c$ of the particle. We use $r_c$ instead of $r_v$ in this equation because the drag force is related more to the cross section of the particle than to its volume. However, we note that using $r_c$ in Eq. 2 is an approximation because

determining the exact Stokes drag force of an irregularly-shaped particle is a more complex issue, see e.g. Loth (2008). Setting





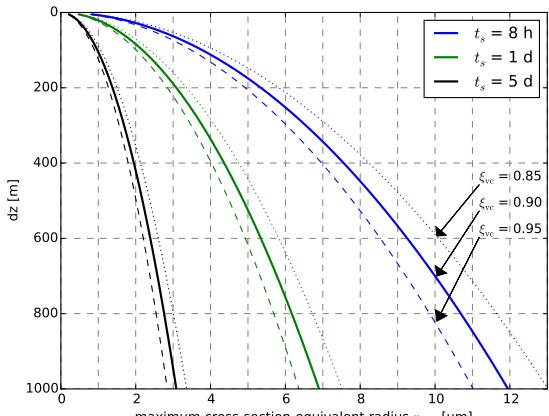

**Figure 1.** Maximum cross-section-equivalent radius $r_{\max}$ of dust particles as function of distance $dz$ from the SAL top after different settling time periods $t_s$ assuming different shape-dependent conversion factors $\xi_{\mathrm{vc}}$.

$F_g = F_d$ and using the conversion factor $\xi_{\mathrm{vc}} = r_v/r_c$ (Gasteiger et al., 2011) results in a settling velocity of the particle in still air of

$$v = \frac{2g\rho}{9\eta} \cdot r_c^2 \cdot \xi_{\mathrm{vc}}^3 \qquad (3)$$

$\xi_{\mathrm{vc}}$ of our six irregular model shapes are 0.955 (shape A), 0.932 (B), 0.911 (C), 0.871 (D), 0.925 (E), and 0.866 (F). Note that
the dynamic shape factor $\chi$ (Hinds, 1999) is $\chi = \xi_{\mathrm{vc}}^{-1}$ if $r_v$ is assumed, and $\chi = \xi_{\mathrm{vc}}^{-3}$ if $r_c$ is assumed for the radius. Henceforth
unless otherwise stated, we use the cross-section-equivalent radius $r = r_c$ for describing particle size.

As a result of gravitational settling during a time period $t_s$ without convective mixing, the maximum particle radius $r_{\max}$ at
a distance $dz$ from the upper boundary of the SAL is given by

$$r_{\max} = \sqrt{\frac{9\eta dz}{2g\rho\xi_{\mathrm{vc}}^3 t_s}} \qquad (4)$$

Fig. 1 illustrates $r_{\max}$ as function of $dz$ for different settling time periods $t_s$. The solid lines show $r_{\max}$ for a conversion factor
$\xi_{\mathrm{vc}} = 0.9$. The vertical axis was chosen such that the top of the SAL ($dz = 0$ m) is at the top of the figure. For example, at $t_s =$
5 d, no particles with radii $r > 2\,\mu\mathrm{m}$ and $\xi_{\mathrm{vc}} = 0.9$ exist in the upper 400 m of the SAL.

### 2.2 Hypotheses about occurrence of convective mixing

In our first hypothesis (H1) we assume that no convective mixing occurs in the SAL over the Atlantic. By contrast, in our
second hypothesis (H2) we assume a diurnal cycle of the convection activity. The idea behind H2 is that the SAL is heated
by absorption of sunlight by the aerosol particles triggering convective mixing during the day. The possibility of convective
mixing in the SAL is consistent with the mostly height-independent potential temperature profiles observed within the SAL





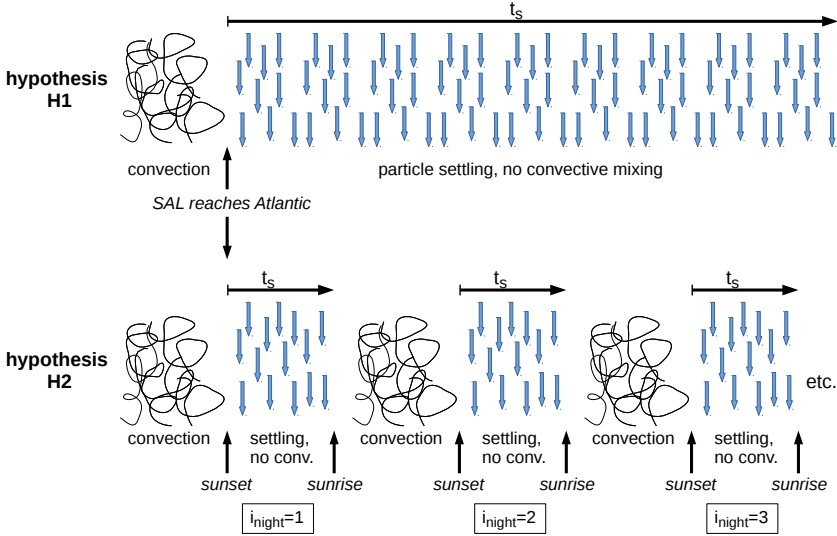

**Figure 2.** Schematic view of hypotheses about the occurrence of convective mixing within the SAL.

(see e.g. Carlson and Prospero (1972) or further down on Fig. 8). We emphasize that the convection activity studied here is not connected to the convection occurring in the marine boundary layer (which sometimes affects also the lower-most parts of the SAL).

Both hypotheses are illustrated in Fig. 2. For simplicity, we assume that the SAL reaches the Atlantic at first sunset. Further-
more, we assume convective mixing always to be perfect though in reality convection may be weak and the mixing imperfect. In case of H2, the initial aerosol size distribution at $t_s = 0$ varies from night to night because a certain fraction of particles is removed by settling during the convection-free time each night before convective mixing starts again with sunrise. The fraction $f$ of particles with radius $r$ that is removed from the SAL each night is calculated for $z_{\mathrm{fallen}}(r) < H_{\mathrm{SAL}}$ with

$$f(r) = \frac{exp\left(\frac{H_{\mathrm{SAL}}-z_{\mathrm{fallen}}(r)}{H_{\mathrm{scale}}}\right) - 1}{exp\left(\frac{H_{\mathrm{SAL}}}{H_{\mathrm{scale}}}\right) - 1} \tag{5}$$

where $z_{\mathrm{fallen}}(r)$ is the distance the particles have fallen (with $v$ as given by Eq. 3) during the night. The duration of the night is set to 11 h in our model, which is a typical value for the northern tropical Atlantic during summertime. $H_{\mathrm{SAL}}$ is the depth of the SAL within which convective mixing occurs each day. We use $H_{\mathrm{SAL}} = 3$ km. To consider the height dependence of the particle concentration present in case of well-mixed layers (as a result of the decrease of air density with height), we assume a scale height $H_{\mathrm{scale}} = 10$ km.

Figure 3 shows for H2 the fraction of the particles present in the SAL at the beginning of each night (counted by $i_{\mathrm{night}}$ as illustrated in Fig. 2). This fraction is calculated as

$$(1 - f(r))^{(i_{\mathrm{night}}-1)} \tag{6}$$





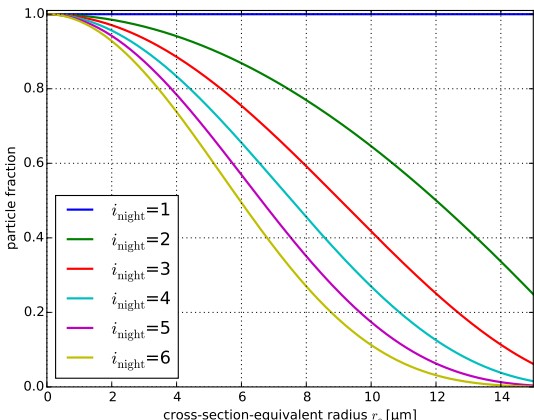

**Figure 3.** Fraction of particles existing in SAL at beginning of each night in case of H2 and $\xi_{vc} = 0.90$.

Henceforward in this paper, we denote the hypotheses and points of time using the following notation: For the first hypothesis we write [H1, $t_s$] and for the second hypothesis we write [H2, $i_{\mathrm{night}}$, $t_s$].

### 2.3 Aerosol mixtures and optical modeling

We simulate the optical properties of Saharan aerosols as described by Gasteiger et al. (2011) and use the reference ensemble

described in that contribution. The lidar-related optical properties of this ensemble are consistent with measurements of Saharan aerosols over Africa (Gasteiger et al., 2011). The aerosol ensembles consist of four log-normal size modes according to the 'desert mixture' of OPAC (Hess et al., 1998) with one mode of small spherical water-soluble (WASO) particles at 50% r.h. ($\rho = 1.42 \cdot 10^3$ kg m$^{-3}$) and three modes of non-spherical mineral dust particles. Mixing of mineral dust with WASO particles is consistent with results presented by Kaaden et al. (2009) who identified Saharan aerosol particles in the smaller size ranges

($r <$ 100-250 nm) to consist mainly of ammonium sulfate. Volatile ammonium sulfate particles were also identified in airborne in-situ measurements (Weinzierl et al., 2009).

The mineral dust particles of the reference ensemble are an equiprobable mix of shapes B, C, D, and F (Gasteiger et al., 2011). The optical properties of dust particles with $2\pi r_v/\lambda \leq 25$ were calculated with the discrete dipole approximation code ADDA (Yurkin and Hoekstra, 2011) and for larger particles it was assumed that the lidar ratio $S$ and the linear depolarization

ratio $\delta_l$ are size-independent, i.e. $S$ and $\delta_l$ calculated for $2\pi r_v/\lambda = 25$ was applied also for larger particles.

It has been shown for Saharan aerosols that the refractive index varies from particle to particle (e.g., Kandler et al., 2011), and that this variability can have significant effects on lidar-relevant optical properties (Gasteiger et al., 2011). In our model, we consider the refractive index variability by the following approximating approach: The imaginary part of the dust refractive index, which is relevant for absorption, is distributed such that 50% of the dust particles are non-absorbing while the other 50%





have an imaginary part that is doubled compared to the value provided by OPAC, leading to good agreement with SAMUM lidar measurements (Gasteiger et al., 2011).

We apply a maximum cut-off radius $r_{\mathrm{max}}$ that is varied as function of distance $dz$ from the SAL top as given by Eq. 4. In case of a diurnal cycle of convective mixing (H2), we consider in addition the partial removal of particles due to settling each night, as described above.

We simulate vertical profiles of the extinction coefficient $\alpha$, the backscatter coefficient $\beta$, the lidar ratio

$$S = \frac{\alpha}{\beta} = \frac{4\pi}{\omega_0 F_{11}(180°)} \tag{7}$$

and the linear depolarization ratio

$$\delta_l = \frac{1 - F_{22}(180°)/F_{11}(180°)}{1 + F_{22}(180°)/F_{11}(180°)} \tag{8}$$

Here, $\omega_0$ is the single scattering albedo of the aerosol particles, $F_{11}(180°)$ and $F_{22}(180°)$ are elements of their scattering matrix at backscattering direction. We consider the height dependence of the particle concentration of the initially well-mixed layer by multiplying all modeled $\alpha$ and $\beta$ profiles with $\exp(dz/H_{\mathrm{scale}})$. In this paper, $\alpha$, $\beta$, $S$, and $\delta_l$ are always aerosol particle properties, i.e. without gas contributions.

## 3 Modeled lidar profiles

In this section we first present modeling results for our first hypothesis (H1) with a settling duration of $t_s = 5$ d, which is the typical time span for transport of aerosol in the SAL from Africa to the Western Atlantic. We investigate the sensitivity of the $\delta_l$ profile shape to the particle shape and the shape-dependent settling velocity. Finally in the last part of this section, we investigate the effect of a diurnal convection cycle (H2) on the $\delta_l$ profile.

### 3.1 Effect of particle settling (H1)

Vertical profiles of lidar-relevant optical properties of the aerosol in the upper 1 km of the SAL, modeled according to H1 after five days without convective mixing ($t_s = 5$ d), are shown in Fig. 4. The solid lines show results for the reference ensemble at three different lidar wavelengths (indicated by color). To illustrate the effect of the WASO particles, we also consider a case in which we removed all WASO particles (dashed lines of same colors).

The lidar ratio $S$ increases towards the top of the SAL (Fig. 4a). $S$ at $\lambda = 532$ nm and 1064 nm has peaks of about 75-80 sr in the upper 70 m of the SAL, decreasing again on the last few meters below the SAL top. Removing the WASO particles from the reference ensemble has a significant effect on $S$ only near the top of the SAL (compare dashed with solid line).

We find a decrease of the linear depolarization ratio $\delta_l$ with height (Fig. 4b). The absolute decrease of $\delta_l$ depends on wavelength; for example, from $dz = 1000$ m to $dz = 100$ m $\delta_l$ decreases by 0.065, 0.074, and 0.121 at $\lambda = 355$ nm, 532 nm, and 1064 nm, respectively. Removing WASO particles strongly increases $\delta_l$ at all heights, in particular at short wavelengths (compare blue lines for $\lambda = 355$ nm). However, the general shape of the $\delta_l$ profiles, i.e. the decrease of $\delta_l$ with height, is independent of whether WASO particles are considered or not.




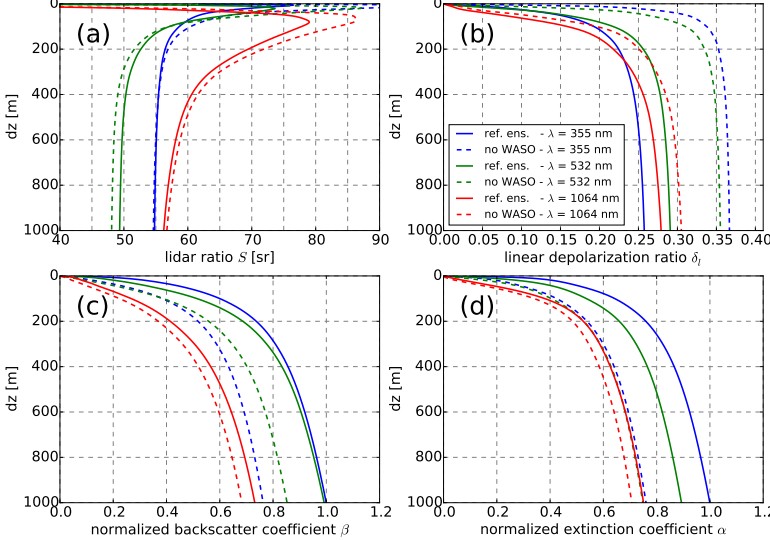

**Figure 4.** Optical aerosol properties of the upper 1 km of the SAL for [H1, 5 d] assuming the reference ensemble (solid lines) at three lidar wavelengths (indicated by color). $\beta$ and $\alpha$ are normalized to the value of the reference ensemble at $\lambda = 355$ nm and $dz = 1000$ m. The dashed lines present profiles when WASO particles are removed from the reference ensemble.

The backscatter coefficient $\beta$, normalized by $\beta$ at $\lambda = 355$ nm and $dz = 1000$ m, is shown in Fig. 4c. It decreases with increasing height; e.g. at $dz = 100$ m and the three lidar wavelengths, $\beta$ of the reference ensemble is reduced by 41%, 49%, and 63%, respectively, compared to $\beta$ at $dz = 1000$ m.

The extinction coefficient $\alpha$ (Fig. 4d) also decreases towards the SAL top; the relative decrease, however, is smaller than for $\beta$, e.g. for the reference ensemble we find values of 36%, 40%, and 50% for the height levels mentioned above. WASO particles influence the wavelength dependence of $\beta$ and $\alpha$ at any $dz$.

### 3.2 Sensitivity of $\delta_l$ profiles to particle shape

The shape mixture in our reference ensemble may not be fully representative for desert aerosol. Therefore, it is worthwhile to estimate the sensitivity of the lidar profiles to particle shape. We focus on $\delta_l$ at $\lambda = 532$ nm because $\delta_l$ is most sensitive to particle shape and many depolarization lidar systems operate at this wavelength.

Fig. 5 shows $\delta_l$ profiles at $\lambda = 532$ nm for [H1, 5 d] where all dust particles of the reference ensemble were replaced by particles of only a single shape, as indicated in the legend. The other properties of the dust particles and the properties of the spherical WASO particles were left unchanged. For each profile, the shape-specific $\xi_{vc}$, as given above, is considered in the calculations.





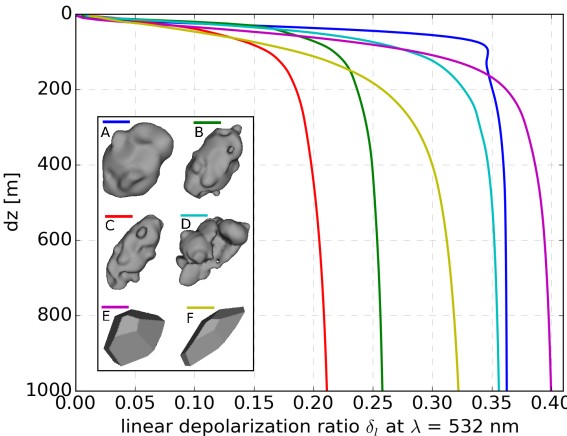

**Figure 5.** Linear depolarization ratio profiles at $\lambda = 532$ nm in the upper 1 km of the SAL for [H1, 5 d]. The reference ensemble is applied (as in Fig. 4), but only a single dust particle shape is assumed in each profile as indicated in the legend.

The absolute value of $\delta_l$ at $\lambda = 532$ nm depends on particle shape with a variation range from about 0.2 to 0.4 at $dz = 1000$ m. $\delta_l$ of elongated shapes (B, C, F) tends to be smaller than $\delta_l$ of the more compact shapes (A, E). The smaller-scale features in the profiles shown in Fig. 5 are a result of the variability of $\delta_l$ as function of size (see Fig. 2 of Gasteiger et al. (2011) where the single particle properties are shown). $\delta_l$ decreases with height for all considered shapes. The decrease of $\delta_l$ in case of shapes D-F (aggregate particles, edged particles) tends to be shifted to lower altitudes compared to $\delta_l$ in case of the other shapes (deformed spheroids).

### 3.3 Sensitivity of $\delta_l$ profiles to shape dependence of settling velocity

As the next step we investigate the importance of the shape dependence of the gravitational settling for $\delta_l$ profiles. For this investigation at least two different dust particle shapes need to be mixed within the model ensembles. Fig. 6 shows $\delta_l$ profiles at $\lambda = 532$ nm for [H1, 5 d] and for three different shape mixtures, which are indicated by color (mixture BCDF corresponds to the reference ensemble). The solid lines illustrate results when shape-dependent $\xi_{vc}$ are considered. By contrast, results shown as dashed lines assumed an average $\xi_{vc}$ value (as indicated in the legend) for the settling of all dust shapes, implying that the settling velocity $v$ varies within the specific ensembles only as function of $r_c$. For comparison also the initial $\delta_l$ profiles ([H1, 0 d]) are shown as dotted lines. Thus, the differences between the dotted and the solid profiles show the total settling effect after 5 days without convective mixing and the differences between solid and dashed lines show the effect that result from the shape dependence of the settling velocity. The latter effect is much smaller than the total settling effect, which allows us to conclude that the shape dependence of the settling velocity is only of minor importance compared to the size dependence. These results are consistent with results presented by Ginoux (2003), but deviate from the model applied by Yang et al. (2013).



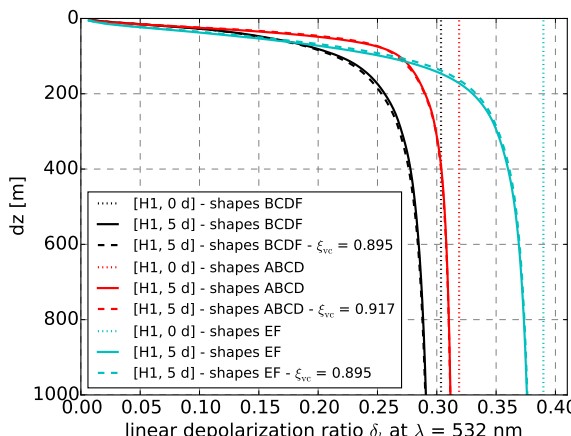

**Figure 6.** Analogous to Fig. 5, but assuming mixtures of different shapes of mineral dust particles as indicated in the legend (shapes BCDF = reference ensemble). The solid lines show profiles for [H1, 5 d]. The dashed lines show the same profiles when no shape dependence of the settling velocity is assumed ($\xi_{vc}$ of all shapes is set to the average value, which is given in the legend). The dotted lines show the modeled profile at the beginning of the settling.

### 3.4 Effect of diurnal convection cycle (H2)

Figure 7 shows $\delta_l$ profiles for both hypotheses, different time periods without convection ($t_s$), and different number of nights ($i_{night}$). The effect of settling on the $\delta_l$ profile increases with increasing convection-free time, in particular in the upper few hundred meters of the SAL (H1, compare dashed lines). In case day-time convective mixing occurs (H2), the night-time $\delta_l$
profile (shown here for $t_s = 8$ h, i.e. 8 hours after sunset) changes only slightly from day to day, with the maximum changes at lower altitudes (compare solid lines). For example, $\delta_l$ is reduced by about 0.007 at $dz = 1000$ m from the first night ($i_{night} = 1$) to the sixth night ($i_{night} = 6$). The difference of the $\delta_l$ profiles between H1 and H2 increases with time (compare lines of same color), illustrating the sensitivity of the $\delta_l$ profiles to the occurrence of convective mixing.

### 4 Comparison with SALTRACE data: Case study 11 July 2013

We now discuss our modeling results based on a comparison with SAL aerosol data measured during the SALTRACE field campaign on 11 July 2013. In this and the subsequent section, we test our two hypotheses.

### 4.1 Lidar measurements

Lidar measurements and radiosonde launches were performed on the grounds of the Caribbean Institute for Meteorology and Hydrology in Bridgetown, Barbados (13.15° N, 59.62° W,  110 m above sea level). Data at $\lambda = 355$ nm and 532 nm from
15 the lidar system POLIS of the LMU (Munich) (Freudenthaler et al., 2016; Groß et al., 2015) and radiosonde data measured





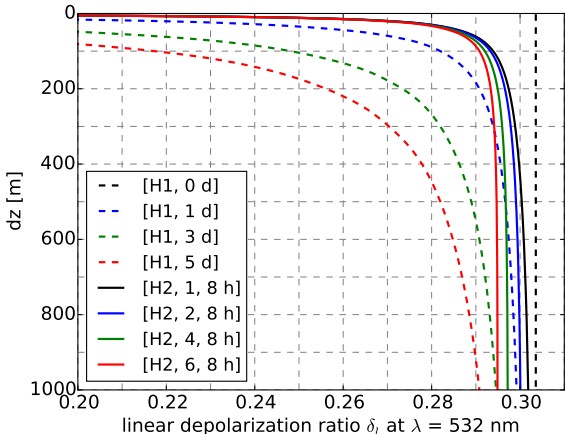

**Figure 7.** Linear depolarization ratio $\delta_l$ profiles at $\lambda = 532$ nm in the upper 1 km of the SAL for both hypotheses after different transport time periods.

by TROPOS (Leipzig) are available. POLIS is a Raman lidar, thus requiring strong temporal and vertical smoothing for the determination of profiles of lidar ratio $S$ and extinction coefficient $\alpha$. A vertical smoothing length of at least 500 m is required for those properties, but even with this smoothing length the signal-to-noise ratio of the Raman measurements is still too low for a meaningful comparison with our modeled vertical profiles. Therefore, we restrict our comparison to the linear depolarization

ratio $\delta_l$ and the backscatter coefficient $\beta$, which are also available from POLIS and require a smoothing length of about 180 m. Furthermore, we consider only $\lambda = 532$ nm for our comparison because this wavelength is more sensitive to particle settling effects than $\lambda = 355$ nm (see Fig. 4).

The lidar measurements presented in this section were performed around 0-1 UTC on 11 July 2013, which is about 2 h after sunset. Back trajectory analysis for this air mass using HYSPLIT (Draxler and Rolph, 2015) suggests that it had left the African

continent about 5 days before the measurements (not shown). Thus, to test our hypotheses about the occurrence of convective mixing, we assume for this comparison $t_s = 2$ h and $i_{\mathrm{night}} = 6$ in case of H2, and $t_s = 5$ d in case of H1.

Fig. 8 illustrates measured and modeled vertical profiles. Fig. 8a shows radiosonde data of water vapour mixing ratio (magenta) and potential temperature (black). The potential temperature is nearly constant within the SAL, which extends up to about 4600 m above ground. This potential temperature profile indicates that convection could have occurred during the trans-

port of this air mass over the Atlantic. The relative humidity in 4500 - 4600 m is about 50% to 54%. The vertical structure of the water vapour mixing ratio and the potential temperature might be regarded as typical for the Saharan Air Layer (Carlson and Prospero, 1972). Fig. 8b contains measured particle backscatter coefficients $\beta$ (black line), as well as profiles calculated for our hypotheses H1 (red) and H2 (green). The amount of particles in the model was scaled to match the measured $\beta$ close to the top of the SAL. Fig. 8c shows corresponding measured and modeled $\delta_l$ profiles. The top of the SAL was set to 4660 m

in the modeled profiles. For [H1, 5 d] we considered, in addition, a case with the SAL top height set to 4740 m which better





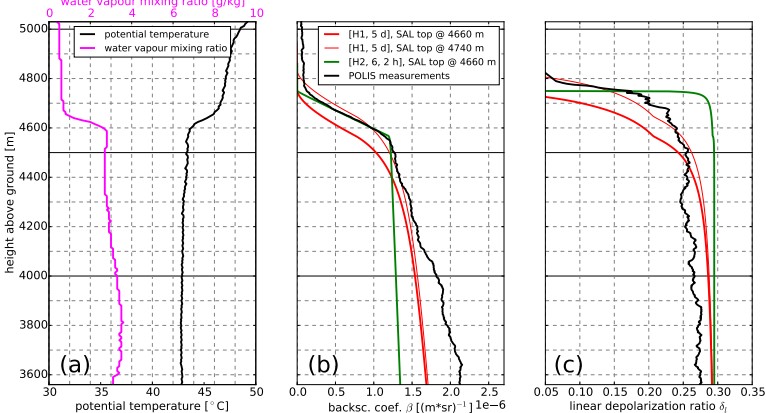

**Figure 8.** Vertical profiles over Barbados around 0 UTC on 11 July 2013. (a) Profile of potential temperature (black) and water vapour mixing ratio (magenta) from a radiosonde launched at 23:39 UTC on 10 July 2013. (b) Particle backscatter coefficients $\beta$ at $\lambda = 532$ nm. (c) Particle linear depolarization ratios $\delta_l$ at $\lambda = 532$ nm. Both $\beta$ and $\delta_l$ measured with POLIS between 0:00 UTC and 0:57 UTC (black). Corresponding modeled profiles assuming a SAL top height of 4660 m, as well as H1 (red) and H2 (green). For H1 also a profile for a SAL top height of 4740 m is plotted (thin red lines). A flat smoothing window of 180 m is used for the measured and modeled lidar profiles.

matches the lidar measurements (thin red line). Measured and modeled lidar profiles shown in Fig. 8 were vertically smoothed with a flat window of about 180 m length.

Modelled and measured $\beta$ agree fairly well near the SAL top if [H2, 6, 2h] is assumed (Fig. 8b). The slope of $\beta$ near the SAL top is mainly determined by the smoothing length applied. Below 4400 m the measured $\beta$ is larger than the modeled
one, which hints to inhomogeneities in the spatial aerosol distributions not considered in our model. In case of [H1, 5 d] (thick red line) deviations from $\beta$ measurements are large, but they are reduced if the SAL top height is 4740 m (thin red line). This shift in top height reduces the consistency with the radiosonde data; we cannot exclude that such differences in SAL top height between radiosonde and lidar measurements are real, but we have no further evidence for that.

The modeled $\delta_l$ in the SAL is generally larger than the measured $\delta_l$ (Fig. 8c), which may be due to the natural variability
between different aerosol source regions. While assuming H2 (green) better captures the measured steep decrease near the SAL top, assuming H1 with a higher SAL top height than found in the radiosonde data (thin red line) somewhat better captures the shape of the $\delta_l$ profile in the upper 1 km. If we assume that the SAL top height shift between lidar and radiosonde could be real, we can conclude that both hypotheses roughly explain the measurements and that uncertainties in our model and the measurements do not allow us to determine which of our hypotheses fits better. However, if we assume that the SAL top height
did not vary that strong between radiosonde and lidar measurements, H2 (convection over Atlantic) fits better than H1 (no convection).



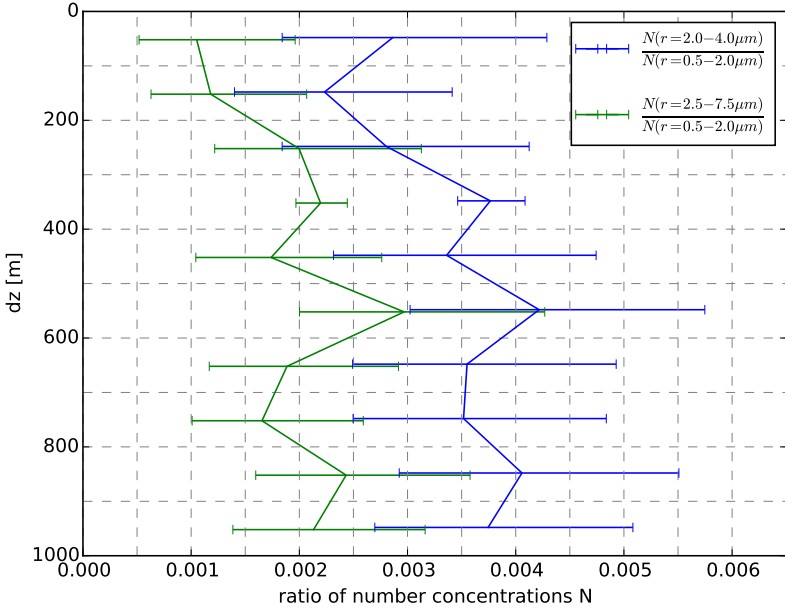

**Figure 9.** Ratios between number concentrations $N$ in different radius ranges measured by CAS-DPOL during Falcon ascent and descent between 12 and 15 UTC on 11 July 2013 close to Barbados (day-time). The vertical axis shows the distance from the SAL top. The data was grouped in 100 m wide bins. The error-bars are Poisson 95% confidence intervals.

## 4.2 Optical particle counter measurements

During SALTRACE, the optical particle counter instrument CAS-DPOL (DMT, Boulder, CO, USA) was operated under the wings of the Falcon aircraft. The ambient air flows passively through the instrument. It has a laser as light source operating at $\lambda = 658$ nm and measures the intensity of light scattered forward to 4°-12° by individual particles flying through its sampling area. Each particle is counted and from the measured intensity its size is inverted. The counts are collected in size bins, covering a nominal radius range from 0.25 μm to 25 μm. From these counts, and knowing the true airspeed of the aircraft and the size of the sampling area, the ambient particle number concentration $N$ is calculated for each size bin. Thereby, the CAS-DPOL provides in-situ measurements of the coarse mode size distribution allowing us to test more directly the size distributions resulting from our hypotheses H1 and H2.

For this test we use data from a flight performed during day-time on 11 July 2013. Using the nominal size bins provided by the manufacturer of the instrument, the data was grouped in size ranges that are affected differently by particle settling. Fig. 9 shows ratios of number concentrations $N$: The blue graph shows the ratio between $N$ in the range $r = 2.0$ μm to 4.0 μm and $N$ in the range $r = 0.5$ μm to 2.0 μm, whereas the green line shows the ratio between $N$ in the range $r = 2.5$ μm to 7.5 μm and $N$ in the range $r = 0.5$ μm to 2.0 μm. Because of the strong decrease of $dN/dr$ with increasing size of coarse particles, $N$ in





these size ranges is mainly determined by $dN/dr$ at the lower boundaries of the ranges. Therefore, the plotted ratios basically are ratios between the number of coarse particles (potentially affected by particle settling) and the number of smaller particles (hardly affected by particle settling). The in-situ data is available at one second resolution, which corresponds to about 8 m vertical resolution during ascent and descent of the investigated flight. Because of the low number of counts per second for $r$

$\geq 2.0$ µm, we grouped the data in 100 m wide vertical bins. The vertical bins are described by their distance $dz$ from the SAL top, with the SAL top at 4630 m asl during ascent and at 4550 m asl during descent as determined from the CAS-DPOL data. Each $dz$ bin covers about 25 seconds of data, except the bin from $dz = 300$ m to 400 m, which covers about 500 seconds of data because the descent was paused flying at constant altitude for several minutes. As a result of the vertical binning, the number of detected particles with $r = 2.5$ µm to 7.5 µm is increased to 14 - 24 in each bin at $dz \geq 400$ m.

As illustrated in Fig. 1, we expect in case of our first hypothesis (H1) that particles larger than $r \approx 2.5$ µm are removed from $dz < 600$ m after 5 days over the Atlantic. However, the green line in Fig. 9 shows that such particles are detected also in the upper 100 m of the SAL. This indicates H1 to be unrealistic even if intrinsic uncertainties of the size determination by CAS-DPOL on the order of $\pm 50\%$ are assumed. In case of H2, we assume that the SAL is well-mixed and thus the aerosol profiles are expected to be height-independent during day-time. Though a reduction of the fraction of large particles towards the SAL

top is indicated in the measured in-situ profiles (Fig. 9), H2 agrees much better with the in-situ data than H1, suggesting that some processes within the SAL keeps large particles longer in the air than expected from gravitational settling.

## 5   Comparison with average $\delta_l$ profiles from POLIS and CALIOP

After presenting the SALTRACE case study in the previous section, we now use averaged lidar profiles to test our hypotheses. Therefore, we first describe how we average the POLIS and CALIOP $\delta_l$ data and then compare our modeling results to these

averaged profiles in the upper 1 km of the SAL.

### 5.1   Averaging POLIS data

We consider the average $\delta_l$ profile at $\lambda = 532$ nm from POLIS during the SALTRACE measurement period from 20 June 2013 to 13 July 2013 at Barbados to get a more robust characterization of SAL profiles after long-range transport. The measurements used for averaging were performed during day and night. Sequences where SAL aerosol was measured were selected by visual

inspection of the $\beta$ and $\delta_l$ profiles. 27 measurement sequences covering about 32 hours were considered for averaging. Only few sequences were performed in the hours before sunrise and around noon time. For the evaluation of the lidar data used in this section, a flat smoothing window of about 40 m and a height-independent lidar ratio of 55 sr is applied to the measurements. Averaging was performed relative to the SAL top, which was localized for each sequence also by visual inspection of the profiles. The average depolarization ratio was calculated from the summed up backscatter coefficients that had been separated

into their parallel and perpendicular components.



## 5.2 Averaging CALIOP data

To get a more general view on the optical properties of the aerosols at the top of the SAL during the transport from Africa over the Atlantic we evaluate measurements of the CALIOP lidar. We restrict our analysis again to $\delta_l$ at 532 nm as this parameter is relatively insensitive to errors encountered in the extinction-backscatter retrieval (Liu et al., 2013), which may result, e.g.

from uncertainties in the lidar ratio (Wandinger et al., 2010; Amiridis et al., 2013). We again analyze the upper 1 km of the SAL, where potential settling effects would be observable with lidar (Fig. 7). We use CALIPSO level 2 aerosol profile products v3.01 (NASA, 2010) of backscatter coefficients $\beta$ and the perpendicular components of the backscatter coefficients $\beta_\perp$ at $\lambda =$ 532 nm measured during summer 2007-2011, i.e. from June to August of each of the five years. We excluded profiles measured on 2 Aug 2009 because of unrealistic large measurement uncertainties found on that day. Powell et al. (2009) describe how

backscattering quantities are calculated from the CALIOP raw data. Vaughan et al. (2009) show the automated procedure to detect aerosol and cloud layers from these backscattering quantities, and Liu et al. (2009) demonstrate how aerosols are discriminated from clouds. We restrict our evaluation to night-time measurements in the region from 10°N to 30°N and 0°W to 80°W. We group these measurements in four longitude ranges of 20° width along the transport path from Africa to the Western Atlantic. All measurements were performed approximately 8 h after sunset.

The CALIOP measurements are performed with a vertical resolution of 30 m and a horizontal resolution of 330 m. The backscatter coefficients $\beta$ and $\beta_\perp$ are provided in the level 2 data with a vertical resolution of 60 m (i.e. for bins of 60 m height) and a horizontal resolution of 5 km, which reduces the noise compared to the measured resolution. As discussed by Vaughan et al. (2009), aerosol features are detected with 30 m vertical resolution using an iterative procedure starting with the horizontal resolution of 5 km. As the noise can be considerable at 5 km resolution, in particular if particle concentrations

are low, the horizontal resolution is subsequently increased to 20 km and 80 km to detect also weaker features. Depending on the results of the feature detection, the backscatter coefficients are horizontally averaged over 5 km, 20 km, or 80 km, and the horizontal averaging range can depend on height. In the following we use only data horizontally averaged over 5 km.

From the large set of aerosol profiles, the profiles that fulfill the following criteria are selected for averaging:

– Uppermost aerosol-containing bin between 3 and 8 km above sea level

– Both sub-bins (30 m resolution) of uppermost aerosol-containing bin classified as aerosol-containing

– All 16 bins (i.e. up to ≈1 km) below uppermost aerosol-containing bin also classified as aerosol-containing

– No cloud-containing bin detected in or above 17 uppermost aerosol-containing bins

– Data horizontally averaged over 5 km (not 20 or 80 km) in each of the 17 uppermost aerosol-containing bins

– Linear depolarization ratio $\delta_l$, averaged over 17 uppermost aerosol-containing bins, larger than 0.10





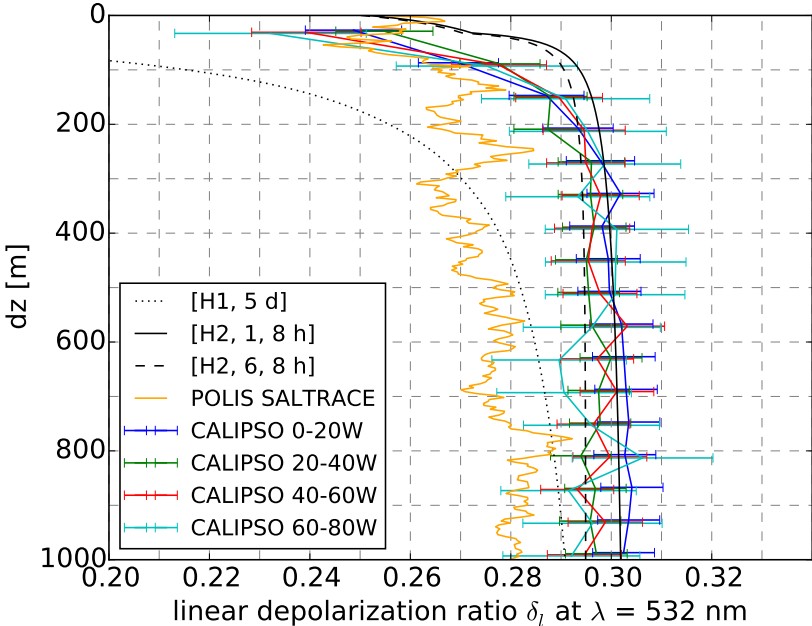

**Figure 10.** Linear depolarization ratio $\delta_l$ profiles at $\lambda = 532$ nm in the upper 1 km of the SAL. The average profile measured by POLIS during SALTRACE in Barbados is shown as the orange line. CALIOP profiles averaged over profiles from summer months 2007-2011 that fulfill conditions listed in the text are shown as dark blue, green, red, and light blue lines. These colors denote different longitude ranges. Error bars of the CALIOP profiles show the statistical uncertainty of the mean. For comparison also model results for different setups are shown as black lines. A flat smoothing window of 60 m is used for the modeled lidar profiles.

After the selection of profiles, $\beta$ and $\beta_\perp$ of each of the 17 vertical bins is summed up over all selected profiles. From these sums, the average $\delta_l$ for each bin is calculated according to

$$\delta_l = \frac{\beta_\perp}{\beta - \beta_\perp} \tag{9}$$

The measurement uncertainties of $\beta$ and $\beta_\perp$ provided in the CALIOP profile data are assumed to be random and uncorrelated (Young, 2010). They are mainly determined by the signal-to-noise ratio (Hunt et al., 2009). As we average over a large number of profiles, the relative uncertainties of the averaged profiles are reduced considerably compared to single profiles.

### 5.3 Comparison with averaged profiles

The averaged linear depolarization ratio profile from POLIS decreases slightly with increasing altitude from about 0.28 at $dz = 1$ km to about 0.27 at $dz = 100$ m and to about 0.25-0.26 for $dz < 100$ m (see orange line in Fig. 10). No significant differences in the average profiles were found between day-time and night-time data (not shown).





Figure 10 shows also the averaged $\delta_l$ profiles calculated from the CALIOP profile data considering all profiles that fulfilled the above-mentioned criteria. The averaged data of the uppermost bin is plotted at $dz \approx 30$ m, the subsequent bin at $dz \approx 90$ m, etc. While the dark blue line shows the average $\delta_l$ close to the aerosol source region, the distance from the source region increases for the green, red, and light blue lines. The light blue line shows the average from a region mainly covering the

Caribbean. Averages were taken over 7623, 7502, 5459, and 1738 individual profiles in the 4 respective regions along the SAL transport path. Considering the statistical uncertainty of the mean, the mean $\delta_l$ does not vary along the SAL transport path and is height-independent with values close to 0.30 for $dz > 250$ m. $\delta_l$ decreases towards the SAL top to values of about 0.23-0.26 in the uppermost bin.

There are some deviations between the averaged profile from POLIS and from CALIOP (Fig. 10). These deviations can be a

result of the natural variability of the desert aerosol properties in combination with the differences in the temporal and spatial coverage of these data sets. In addition, the assumptions made about the lidar ratio potentially affect the POLIS data shown in Fig. 10.

Comparing the measured with the modeled profiles, it becomes clear that the strong decrease of $\delta_l$ in the upper 100 m of the SAL, as modeled for long-range transport without convective mixing (H1, black dotted line), is found neither in the averaged

POLIS data nor in the averaged CALIOP data over the Western Atlantic (orange, red, and light blue lines). This indicates our first hypothesis (H1) to be unrealistic. A further result that renders H1 unlikely is the fact that the average $\delta_l$ profile from CALIOP is not modified during transport while one would expect significant changes of the $\delta_l$ profile during transport if H1 is assumed (see dashed lines in Fig. 7).

The black solid and the dashed lines in Fig. 10 show the $\delta_l$ profile when convective mixing is assumed (H2). These modeled

$\delta_l$ profiles are relatively height-independent, except in the upper 100 m of the SAL. Comparing the POLIS measurements, which were averaged over day-time and night-time, with these profiles shows a fairly good agreement in profile shape, but an almost height-independent difference of about 0.01 to 0.02. Despite this difference, this comparison with POLIS data suggests H2 to be much more realistic than H1. Comparison of the CALIOP profiles with our modeling results also suggests that considering convective mixing (H2) is required to explain the measured data sets.

Our model for H2 predicts a reduction of $\delta_l$ by about 0.007 at $dz = 1$ km after about 5 days (Fig. 7). This reduction in not seen in the CALIOP profiles, possibly because it is within the range of the statistical uncertainty of the averaged $\delta_l$ profiles from CALIOP (Fig. 10).

### 5.4 Discussion of comparison between H2 and CALIOP profiles

Our model assuming day-time convection (H2) agrees well with the averaged CALIOP data. However, deviations of the $\delta_l$

profile between this model and the averaged CALIOP data occur in the upper 2-3 bins (Fig. 10). The measurements indicate stronger removal of large particles near the SAL top over Africa and over the Atlantic than our model for H2 suggests (solid and dashed black lines). There are several potential reasons for deviations: In our model we assume perfect mixing of particles over the complete SAL when convection occurs and we assume a sharp boundary between the SAL and the layer above. But it is plausible that some mixing occurs between the SAL and the layer above on the order of 10 m. Thus, some desert aerosol near





the upper boundary of the SAL might be decoupled from the day-time convection within the SAL, allowing a stronger removal of large particles than our model predicts. However, one fact that contradicts these considerations is that the $\delta_l$ profile seems to be independent of transport time (compare CALIOP profiles in Fig. 10) while one would expect that such effects become larger with transport time.

A further aspect that has to be kept in mind is that multiple scattering could affect the CALIOP measurements (Wandinger et al., 2010). In case of our SAL top study, the multiple scattering effect would increase with increasing $dz$ as the lidar pulse penetrates the SAL. It is well-known that with increasing amount of particles the multiple scattering effect increases (e.g. Bissonnette et al., 1995). Using the CALIOP profile data we do not find a significant dependence of the $\delta_l$ profiles on the absolute values of $\beta$ (not shown), indicating that multiple scattering does not significantly affect the averaged $\delta_l$ profiles.

To investigate the CALIOP profiles in more detail, an analysis is provided in supplement S1, considering also 20 km and 80 km horizontal averages, the year-by-year variability, sub-bin classification (which is provided in the profile data at 30 m vertical resolution), and the sensitivity to the $\delta_l$ threshold. The sensitivity of the $\delta_l$ profile shapes to these parameters was found to be low. As a consequence, it seems likely that the simplifications in our model are the reason for the remaining deviations near the SAL top.

## 6  Summary and conclusions

Transport of aerosol in the Saharan Air Layer (SAL) over the Atlantic is relevant for weather and climate but important processes within the SAL still are not well understood. To gain insights in relevant processes, we developed a model which describes the modification of the vertical aerosol distribution in the upper 1 km of the SAL during transport based on the physical processes of gravitational settling and convective mixing. From the vertical aerosol distributions, lidar profiles are calculated using explicit optical modeling. Sensitivity studies revealed (a) that generally the particle linear depolarization ratio decreases towards the SAL top for all considered model shapes, and (b) that the size dependence of the settling velocity is significantly more important for the linear depolarization ratio profile than its shape dependence.

The model results were compared to lidar and in-situ measurements and two hypotheses about the occurrence of convective mixing within the SAL were tested. Comparisons with measurements in Barbados, performed in the frame of the SALTRACE campaign, indicate that it is very likely that convective mixing occurs in the SAL over the Atlantic. These findings are strongly supported by an analysis using night-time depolarization profiles from CALIOP. Furthermore, the CALIOP data shows that the average linear depolarization ratio profile in the upper 1 km of the SAL does not change along its transport path over the Atlantic. Our comparisons lead us to the conclusion that the size distributions, at least in the range $r <\approx$ 3-5 μm, are hardly modified during transport after the first night after mobilization. These findings are consistent with results from other studies about Saharan aerosol transport, e.g. Reid et al. (2003); Maring et al. (2003); Weinzierl et al. (2011); Mahowald et al. (2014); Denjean et al. (2016); Weinzierl et al. (2016).

Our model assuming day-time convective mixing, which is driven by the idea that the Saharan aerosol absorbs sunlight triggering convection, explains the CALIOP data quite well. However some deviations remain that might originate from the



simplifications of our model. For example, we did not consider the possibility of night-time convection, of weak convection (i.e. non-perfect vertical mixing), of size-selective particle removal at the lower boundary of the SAL during convection, or effects due to electrical fields in the SAL, which may occur in reality. In this connection, it needs to be investigated if heating by solar radiation during day-time is indeed the main driving force for convection. For example, also radiative effects in

the thermal infrared could be an important aspect to consider for convective mixing, as discussed by Carlson and Prospero (1972). Measurements of aerosol properties and vertical wind speeds with high vertical and temporal resolution using lidar in combination with airborne in-situ instruments, radiative transfer calculations, and convective models might help to deepen our understanding of processes affecting size distributions and lifetime of super-micron particles.

*Acknowledgements.* The research leading to these results has received funding from LMU Munich's Institutional Strategy LMUexcellent

within the framework of the German Excellence Initiative and from the European Research Council under the European Community's Horizon 2020 research and innovation framework program/ERC grant agreement No. 640458 - A-LIFE. S. Groß acknowledges funding by a DLR VO-R young investigator group. The SALTRACE campaign was mainly funded by the Helmholtz Association, DLR, LMU, and TROPOS. We thank the ground-based remote sensing group of the Leibniz-Institut für Troposphärenforschung (TROPOS), led by Albert Ansmann, for providing radiosonde data. The Caribbean Institute for Meteorology and Hydrology in Bridgetown, Barbados kindly provided

the infrastructure to perform the SALTRACE lidar measurements. The CALIOP data were obtained from the NASA Langley Research Center Atmospheric Science Data Center. We thank the reviewers of a previous version of this manuscript for suggestions that helped us to substantially improve it.





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
