# Peer review of "Particle settling and vertical mixing in the Saharan Air Layer as seen from an integrated model, lidar, and in-situ perspective"

_Atmospheric Chemistry and Physics, 2016_

## Referee Comment (RC1) · Anonymous Referee #1 · 21 Jul 2016

The paper tests two hypotheses about convective mixing within the SAL: one that assumes no mixing (H1), and the second that does assume mixing. This second hypothesis (H2) assumes heating due to absorption of sunlight by the dust particles, causing convective mixing during the day, and settling during the night. I think the paper is overall well written, especially the methods and model used are described in detail. However there are some issues that I want to discuss in the review below. The conclusion of this paper is that the scenario as modeled in H1 is unrealistic, leaving H2 to be the most viable option. A real-world scenario can't be as simple as described in H1. However, in many ways H2 also does not seem to be a perfect fit, and in the future perhaps more aspects of the model could be tested, to get insight in what determines

the measured lidar profiles.

Specific comments:

Comments described as page number/line number(s)

2/8: Why is particle size described here as radius? The cited reference (Maring et al., 2003) also uses particle diameter.

3/2-3: Do you have a reference for this data collected during the SALTRACE field campaign?

3/13: Six irregular dust particle shapes, is this enough to accurately represent Saharan dust?

3/24: What is the particle density based on? Is there a reference for this?

3/24-25: What about the drag force of an aerosol particle larger than r = 10 $\mu$m? There are many studies that observe 'giant' dust particles being transported from the Sahara over the Atlantic Ocean (e.g. (Glaccum and Prospero, 1980;Betzer et al., 1988;Goudie and Middleton, 2006;Mahowald et al., 2014;Middleton et al., 2001;Kok, 2011)).

5/Figure 2: Maybe this figure can be mirrored, so that it resembles the E-W transport over the Atlantic Ocean, as is described in the paper.

5/12: HSAL also decreases westward (Adams et al., 2012), which may affect the modeled results? (As also shown in in the supplementary data, Figure S-11).

6/Figure 3: At inight = 0, the particle concentration of every size class is 1. Is this realistic, since particles are never distributed equally in any given sample? (e.g. (Stuut et al., 2005)). And are these number concentrations or volume/surface distributions?

6/6-7: Can you show the grain-size distributions that you have used in your model? What is the maximum grain size used?

7/15-16: Do you have a reference for this? E.g. Prospero (1996), Schütz (1980).

[Figure]

8/8-9: This may also be true for other aspects, like chemistry (and refractive index) and particle size. In how far have you tested how representative that data is for realistic cases?

9/14-17: Yes, there seems a minor effect when the shape conversion factor is taken as an average, but there is a substantial difference between the different shape mixtures (different colors in Fig. 6). What shape mixture was eventually used for the comparison with POLIS and CALIOP data?

10/10-11: Is there a reference that describes this field campaign and its data?

11/9-10: Why are the backward trajectories not shown? At what heights were the air layers modeled? Maybe satellite imagery can help to track the dust layer? (It seems like the dust was emitted at July 4th) see: https://worldview.earthdata.nasa.gov/

11/18-19: Isn't the particle concentration measured as well? (Section 4.2, page 13) Can't that be used as input for the model?

11/19-20: Why was the top of the SAL set at exactly 4660 m? What was this height based on? Based on the potential temperature it is said to be at 4600 m (11/13-14).

11/20: Why is a SAL height of 4740 m modeled? Apparently it matches the data better, but the data implies the top of the SAL may be at 4600 m (11/13-14), so maybe other parameters have to be adjusted to let the model better fit the data?

12/3-8: It seems that for the lower atmosphere, H1 fits better with the measurements, for both $\beta$ and $\delta l$ (Fig. 8b and 8c).

12/14-16: In the case of Fig 8, I don't see why H2 would fit better than H1.

13:5-6: This size range of 0.25-25 $\mu$m (radius), is this the same size range as used in the model?

13/8-9: "[...] allowing us to test more directly the size distributions resulting from our hypotheses H1 and H2." Why not do this before the model calculations, and incorporate

the data in the model?

13/10: Again, is there a reference for this aspect of the SALTRACE project that describes the data and how it was obtained? Because there are not many details about this in the current paper (e.g. location, flight plan, etc.)

13/11: "[. . .] the data was grouped in size ranges that are affected differently by particle settling." What is this based on?

13/12-14: Why were these particle sizes considered? Why not the even coarser particles (up to r = 25 $\mu$m)?

14/1-3: Again, where is this effect of size on particle settling based on? What determines these boundaries?

14/9: This seems like a very low amount of particles per size bin, how reproducible are these numbers? Have more of similar flights been performed in this area to compare the data to?

14/22-23: If this data from SALTRACE is available, why not look at more case studies as in section 4? Since every case is unique, this could give more general insights in the processes involved during transportation of dust within the SAL, related to convective mixing. Why was the case of July 11th chosen for comparison with modeled data?

15/5-6: "We again analyze the upper 1 km of the SAL [. . .]", I assume this was also done for averaging the POLIS data, however it is not mentioned in the previous section (5.1).

16/9-10: "No significant differences in the average profiles were found between day-time and night-time data (not shown)." What does this tell us? That there seems to be convection during both night and day? (which is not assumed in your model)

17/19-23: To me it seems that the agreement between the POLIS data and H2 is also not good, just as the bad fit with H1. With the CALIOP data, however, the fit with H2 is

much better.

17/23-24: You seem to discuss the comparison between CALIOP and H2 also in the next paragraph, so maybe this sentence is a bit redundant here.

18/10-14: Could you say a bit more about the supplementary data? If not here, then in the supplement itself?

18/24-25: As mentioned for the previous sections as well, I think the results when comparing the model to POLIS data are not that conclusive, for both the case study and the average POLIS data.

18/28-29: This relation with particle size isn't really mentioned in the discussion before (except for Figure 9).

Technical corrections:

9/8: do you mean "[. . .] the importance of the shape dependence ON the gravitational settling for $\delta$l profiles."?

11/15: "The relative humidity AT 4500 – 4600 m [. . .]"

16/Fig 10 (caption line 4): The statistical uncertainty of the mean, do you mean standard deviation? Also, in the text you refer to CALIOP data, however in the legend now it says CALIPSO, which is not very consistent.

References:

Adams, A. M., Prospero, J. M., and Zhang, C.: CALIPSO-Derived Three-Dimensional Structure of Aerosol over the Atlantic Basin and Adjacent Continents, Journal of Climate, 25, 6862-6879, 10.1175/jcli-d-11-00672.1, 2012. Betzer, P. R., Carder, K. L., Duce, R. A., Merrill, J. T., Tindale, N. W., Uematsu, M., Costello, D. K., Young, R. W., Feely, R. A., Breland, J. A., Bernstein, R. E., and Greco, A. M.: Long-Range Transport of Giant Mineral Aerosol-Particles, Nature, 336, 568-571, 10.1038/336568a0, 1988. Glaccum, R. A., and Prospero, J. M.: Saharan Aerosols over the Tropical North-

Atlantic - Mineralogy, Marine Geology, 37, 295-321, 10.1016/0025-3227(80)90107-3, 1980. Goudie, A. S., and Middleton, N. J.: Desert Dust in the Global System, Springer Berlin Heidelberg New York, 2006. Kok, J. F.: A Scaling Theory for the Size Distribution of Emitted Dust Aerosols Suggests Climate Models Underestimate the Size of the Global Dust Cycle, Proceedings of the National Academy of Sciences, 108, 1016-1021, 10.1073/pnas.1014798108, 2011. Mahowald, N., Albani, S., Kok, J. F., Engelstaeder, S., Scanza, R., Ward, D. S., and Flanner, M. G.: The Size Distribution of Desert Dust Aerosols and its Impact on the Earth System, Aeolian Research, 15, 53-71, http://dx.doi.org/10.1016/j.aeolia.2013.09.002, 2014. Middleton, N. J., Betzer, P. R., and Bull, P. A.: Long-Range Transport of 'Giant' Aeolian Quartz Grains: Linkage with Discrete Sedimentary Sources and Implications for Protective Particle Transfer, Marine Geology, 177, 411-417, 10.1016/s0025-3227(01)00171-2, 2001. Prospero, J. M.: Saharan Dust Transport over the North Atlantic Ocean and Mediterranean: an Overview, in: The impact of desert dust across the Mediterranean, edited by: Guerzoni, S., and Chester, R., Kluwer Academic, Dordrecht / Boston / London, 133-151, 1996. Schütz, L.: Long Range Transport of Desert Dust with Special Emphasis on the Sahara, Annals of the New York Academy of Sciences, 338, 515-532, 10.1111/j.1749-6632.1980.tb17144.x, 1980. Stuut, J. B., Zabel, M., Ratmeyer, V., Helmke, P., Schefuss, E., Lavik, G., and Schneider, R.: Provenance of Present-Day Eolian Dust Collected off NW Africa, Journal of Geophysical Research-Atmospheres, 110, 10.1029/2004jd005161, 2005.

---

## Referee Comment (RC2) · Anonymous Referee #2 · 22 Jul 2016

The authors investigate processes affecting long-range particle transport in the Saharan Air Layer (SAL) by modeling aerosol property profiles for two scenarios: (1) gravitational settling only and (2) gravitational settling alternating with convective mixing in the SAL. Model results are compared with ground- and space-based lidar measurements as well as aircraft particle counter measurements to determine the most likely scenario.

The paper is overall very well written and a valuable contribution that is suitable for publication in ACP. Nevertheless, I have some comments (below).

(1) The authors initialize their model with a reference ensemble. As a result, there is no variation of the modeled profiles for different time periods (see for example Figures S-1 to S-5). Is there any way to initialize the model with measured profiles (for example

using CALIPSO data) and then compute the profile changes based on the assumptions outlined in the model description?

(2) The description of the computation of the fraction of removed particles (p. 5, l. 4-14) is not clear to me. I am not sure how Equation 5 has been derived and why H_scale has been set to 10 km. Some further detail might be beneficial. Also, it is said that f(r) is calculated for z_fallen(r) < H_SAL with H_SAL being the SAL depth. Have the authors considered a starting height of the particles (not all start from the top)? In the context of l. 4 on p. 5, it might also be worth stating that only one air parcel (vertically reaching throughout the SAL) is considered and horizontal mixing is being ignored.

(3) Two hypotheses are considered: (H1) profile changes are caused by gravitational settling only; and (H2) profile changes are caused by gravitational settling during the night and convective mixing during the day. Do I understand correctly, that gravitational settling is being neglected during the day in H2? I would think that gravitational settling occurs always, irrespective of whether or not convective mixing occurs in addition. Looking at Fig. 10, I could imagine that the additional consideration of daytime gravitational settling in H2 might reduce delta_l at large heights (small dz) and thus lead to a "smoother" profile which compares better to the measured profiles.

(4) The cross-sectional particle radius, r_c, is being used for the computation of the drag force (Section 2.1). However, r_c varies with particle orientation. Has this been accounted for in the model? As the authors assume a random particle orientation, the drag force might have to be calculated as an average over the drag force of single particles with different orientation. Is there any information about whether or not particles are being oriented randomly in nature or if the perhaps align with the flow in some way?

(5) The authors suggest that convective mixing in the SAL occurs due to "absorption of sunlight by the aerosol particles". Would this not lead to stronger heating at the top of the SAL compared to the bottom and thus to a stabilization?

(6) Oceanic measurements (e.g. van der Does et al. 2016 [http://www.atmos-chem-

phys-discuss.net/acp-2016-344/]) suggest, that particles of a few tens of microns can still be transported some distance across the Atlantic. Do the authors have any evidence of this from their measurements? If that is the case, then the Stokes regime might not apply anymore (for the largest particles). How would this affect the modeled profiles?

Minor comments:

(1) P. 2, l. 9-10; To me, the statement "cannot explain their measurements with Stokes gravitational settling alone" suggests that Stokes settling might be too weak. However, the subsequent statement, that they had to "reduce the Stokes settling velocity" suggests otherwise. Perhaps rephrasing would clarify this.

(2) P. 3, l. 27; Suggest using half-blanks between unit-parts to avoid Pa (times) s being read as Pas.

(3) P. 4, l.1; Setting $F\_g = F\_d$ suggests that a particle would be suspended and not that it would be in "still air", would it? Unless still air is interpreted such that the particle experiences no vertical movement.

(4) P. 6, l. 10; Suggest using micrometers instead of nanometers for consistency.

(5) P. 7, l. 27; The sentence "We find a decrease of linear depolarization ratio delta_l with height" is somewhat confusing as it is not clear if the authors mean with decreasing or increasing height (or dz). Perhaps reword. This occurs also at other places in the paper (e.g. p. 9, l.4)

(6) P. 9, l. 2; Please clarify which "smaller-scale features" are meant.

(7) P. 9, l.9; Why do two particle shapes need to be mixed within one ensemble? Could the shape-dependence of settling not be investigated using two model ensembles each having only particles of a particular shape? (Perhaps I misunderstood the sentence).

(8) P. 12, l.1; Suggest moving "(thin red line)" to after 4740 m.

(9) P. 17, l.25; is instead of in
* * *

---

## Author Comment (AC1) · 15 Nov 2016

*Reviewer comments are written in italic font*, author's responses in normal font.

**Response to reviewer 1**

*The paper tests two hypotheses about convective mixing within the SAL: one that assumes no mixing (H1), and the second that does assume mixing. This second hypothesis (H2) assumes heating due to absorption of sunlight by the dust particles, causing convective mixing during the day, and settling during the night. I think the paper is*

[Figure]

*overall well written, especially the methods and model used are described in detail. However there are some issues that I want to discuss in the review below. The conclusion of this paper is that the scenario as modeled in H1 is unrealistic, leaving H2 to be the most viable option. A real-world scenario can't be as simple as described in H1. However, in many ways H2 also does not seem to be a perfect fit, and in the future perhaps more aspects of the model could be tested, to get insight in what determines the measured lidar profiles.*

We thank reviewer 1 for his/her thoughtful comments that helped us to improve our manuscript.

*Specific comments:*

*Comments described as page number/line number(s)*

*2/8: Why is particle size described here as radius? The cited reference (Maring et al., 2003) also uses particle diameter.*

The reason is that radius is used more often than diameter in aerosol optical modeling and remote sensing, which is the research field our manuscript originates from. We think, as long as it is clearly stated which size description is used, this should be no issue for the reader.

*3/2-3: Do you have a reference for this data collected during the SALTRACE field campaign?*

We added here the reference to the SALTRACE overview paper (which is in review).

*3/13: Six irregular dust particle shapes, is this enough to accurately represent Saharan dust?*

This number of shapes certainly is not enough to accurately represent all aspects of Saharan dust. In real dust plumes, the number of shapes is 'almost infinite'. Nevertheless, the variability between these six dust-like shapes is large enough to cover the

optical properties of Saharan dust mixtures sufficiently well for the purpose of our study. We added a sentence claiming that using these six shapes is only an approximation.

The issue raised by the reviewer was the idea behind Figure 5. It shows that the linear depolarization ratio decreases towards the SAL top for each of the six very different dust-like shapes.

*3/24: What is the particle density based on? Is there a reference for this?*

The particle density assumption is based on Hess et al. (1998). We added this reference here.

*3/24-25: What about the drag force of an aerosol particle larger than r = 10 $\mu$m? There are many studies that observe 'giant' dust particles being transported from the Sahara over the Atlantic Ocean (e.g. (Glaccum and Prospero, 1980; Betzer et al., 1988; Goudie and Middleton, 2006; Mahowald et al., 2014; Middleton et al., 2001; Kok, 2011)).*

For increasing size the Reynolds number, which is the ratio of inertial to viscous forces, increases (Hinds, 1999). The inertial forces become relevant for $r > 10\ \mu m$, leading to a reduction of the settling velocity compared to Stokes which assumes only viscous forces. However this reduction becomes relevant only slowly: Using the formula given by Hinds, 1999, the Reynolds number for spheres with our assumed dust density of 2600 kg m$^{-3}$ is $\approx$0.04 for r = 10 $\mu$m, $\approx$0.35 for r = 20 $\mu$m, $\approx$1.2 for r = 30 $\mu$m. According to the same book, the error of the drag force calculated by Stokes law is 5 % at Reynolds number 0.3 and 12 % at Reynolds number 1.0. Thus, if we extra-/interpolate between the Reynolds numbers and assume that the proportionality between drag force and velocity (Eq. 2 of discussion paper) is still valid in this size range, we can estimate that the settling velocity is reduced compared to Eq. 3 by about 5 % for r = 20 $\mu$m particles and about 15 % for r = 30 $\mu$m. In case of non-spherical particles things could be a bit more complicated, for example non-random orientation of 'giant' dust particles may lead to a further reduction of the settling velocity (see re-
sponse to reviewer 2). We now briefly discuss this in our article, but prefer not to go in too much detail because such 'giant' dust particles are only of minor importance for our study.

*5/Figure 2: Maybe this figure can be mirrored, so that it resembles the E-W transport over the Atlantic Ocean, as is described in the paper.*

We prefer to keep the direction with increasing time towards the right because our model is based on time spans. To make it a bit easier for the reader to comprehend the relation to geography we added a few geographic names to the figure.

*5/12: HSAL also decreases westward (Adams et al., 2012), which may affect the modeled results? (As also shown in in the supplementary data, Figure S-11).*

The effect of considering a decrease of HSAL (e.g. HSAL = 4 km at the beginning and HSAL = 2 km in the Caribbean) would be that particle removal would be weaker at the beginning and stronger later. Replacing the fixed HSAL of 3 km by this westward decreasing HSAL (4 km before inight=2, 3.5 km before inight=3, etc.) changes $\delta_l$ at $dz$ = 1 km from 0.29490 to 0.29459 for [H2, 6, 8h], and from 0.29854 to 0.29907 for [H2, 3, 8h]. However, in the paper we prefer to use only a single value for simplicity of our idealized model.

*6/Figure 3: At inight = 0, the particle concentration of every size class is 1. Is this realistic, since particles are never distributed equally in any given sample? (e.g. (Stuut et al., 2005)). And are these number concentrations or volume/surface distributions?*

The reviewer probably misinterpreted Fig. 3. Fig. 3 shows the fraction of particles existing in the SAL at the beginning of each night relative to the initial size distribution (reference ensemble described in the subsequent subsection). A fraction of 1 means that no particles had sedimented from the SAL. Thus, at $i_{night}$=1, the unmodified size distribution of the reference ensemble is used. In the subsequent night, for example, ≈50 % of the r = 12 $\mu$m particles are removed from the SAL (see green line in Fig. 3).

This relative fraction could be applied to number, surface, volume distributions. In our case we apply it, following Hess et al. (1998), to dN/dr. We improved the description related to Fig. 3 to make it clearer.

*6/6-7: Can you show the grain-size distributions that you have used in your model? What is the maximum grain size used?*

We now added a figure in the supplement illustrating the number/surface/volume distribution of our reference ensemble, modified for different $i_{night}$. As maximum radius in the reference ensemble we use 40 $\mu$m, which is now mentioned in the paper. During the mixing-free time, the size distribution is cut off at the $r_{max}$ given by Eq. 4.

*7/15-16: Do you have a reference for this? E.g. Prospero (1996), Schütz (1980).*

We now have added reference to Schütz (1980).

*8/8-9: This may also be true for other aspects, like chemistry (and refractive index) and particle size. In how far have you tested how representative that data is for realistic cases?*

The reviewer is correct in that this is certainly true also for refractive index and size distribution because of the quite complex nature of desert aerosol. We use refractive index data and size distribution data that is consistent with measurements near the source regions (and then modify the size distribution as function of transport time and height). So we think that it is 'sufficiently representative' for our study. We tested also the sensitivity of the depolarization ratio profile to realistic variations of refractive index and initial size distributions but we found that the sensitivity is lower than the sensitivity to particle shape (shown in Fig. 5). Now this is mentioned in the text.

*9/14-17: Yes, there seems a minor effect when the shape conversion factor is taken as an average, but there is a substantial difference between the different shape mixtures (different colors in Fig. 6). What shape mixture was eventually used for the comparison with POLIS and CALIOP data?*

We now better describe why we do this comparison. The purpose of showing different shape mixtures is to show that our results with respect to the relative importance of the settling-induced separation of particle size versus the separation of particle shape is not specific to the reference ensemble. For the comparison with the measurements we use mixture BCDF (reference ensemble, shown as black lines), which was found to provide optical properties consistent with lidar measurements near the Sahara (Gasteiger et al., 2011).

*10/10-11: Is there a reference that describes this field campaign and its data?*

We added reference to the SALTRACE overview paper.

*11/9-10: Why are the backward trajectories not shown? At what heights were the air layers modeled? Maybe satellite imagery can help to track the dust layer? (It seems like the dust was emitted at July 4th) see: https://worldview.earthdata.nasa.gov/*

We now added backtrajectories to the supplement. As endpoint height we took the heights where the SAL was detected over Barbados (3.6, 4.0, 4.4 km in the plot added to the supplement). Emission on July 4th seems plausible to us. In H1 we assume that settling starts only when the SAL reaches the Atlantic, thus in Sect. 4 we need to look at this later point of time. If we would consider settling over Africa as well (meaning that we increase $t_s$ in our model from 5 days to about 7 days) the depolarization curves for H1 would be shifted to larger $dz$ (by a factor of 7/5). However, that would not change the conclusions of our paper.

*11/18-19: Isn't the particle concentration measured as well? (Section 4.2, page 13) Can't that be used as input for the model?*

Yes, the particle concentration was measured as well, however about 13 hours later. As the amount of particles strongly varies with time and location, these measurements can not be used as input here.

*11/19-20: Why was the top of the SAL set at exactly 4660 m? What was this height*

*based on? Based on the potential temperature it is said to be at 4600 m (11/13-14).*

As we do have some spatial and temporal distance between the radiosonde and the lidar measurements, we assumed that the SAL top height could vary between lidar and radiosonde. The 4660 m was used because it best fits the H2 model to the measurements.

*11/20: Why is a SAL height of 4740 m modeled? Apparently it matches the data better, but the data implies the top of the SAL may be at 4600 m (11/13-14), so maybe other parameters have to be adjusted to let the model better fit the data?*

4740 m was selected to fit the H1 model to the measurements. Modifying shape, refractive index, size distribution does not help much to reduce the discrepancy between the SAL top height fitted to the POLIS data in case of H1 and the SAL top height measured by the radiosonde.

*12/3-8: It seems that for the lower atmosphere, H1 fits better with the measurements, for both $\beta$ and $\delta_l$ (Fig. 8b and 8c).*

We agree. However, the upper part of the SAL is more sensitive to the discussed settling effects. In case of $\beta$ we have to keep in mind that this is an extensive parameter, depending on the variable amount of particles. We now have reevalutated the POLIS data using a reduced vertical smoothing range. The $\delta_l$ profile lies between H1 and H2 and the uncertainties of the measurements are in the same range as the differences between H1 and H2.

*12/14-16: In the case of Fig 8, I don't see why H2 would fit better than H1.*

Since we have no further information about the exact SAL top height during the lidar measurements, we now discuss this comparison saying that the lidar data does not fit better to one of our hypotheses.

*13/5-6: This size range of 0.25-25 $\mu$m (radius), is this the same size range as used in the model?*

No, this is a different size range. However, both size ranges cover the sizes of interest (see Fig. 1+2 and the new Fig. S-1 in the supplement).

*13/8-9: '[. . .] allowing us to test more directly the size distributions resulting from our hypotheses H1 and H2.' Why not do this before the model calculations, and incorporate the data in the model?*

In our current model it is not possible to incorporate the data collected over Barbados because our model needs the aerosol mixture at the start of the transport (i.e. data collected over Africa). Of course, we could show the in-situ data before the lidar data. But as both data sets are independent of each other, this would not change our conclusions.

*13/10: Again, is there a reference for this aspect of the SALTRACE project that describes the data and how it was obtained? Because there are not many details about this in the current paper (e.g. location, flight plan, etc.)*

More information is given in the SALTRACE overview paper. In addition, we added in the supplement a map where the locations of the measurements are illustrated.

*13/11: '[. . .] the data was grouped in size ranges that are affected differently by particle settling.' What is this based on?*

The nominal size bins are provided by the manufacturer of CAS-DPOL and the size ranges were selected based on Eq. 3 / Fig. 1 such that we expect a significant height dependence if H1 is true.

*13/12-14: Why were these particle sizes considered? Why not the even coarser particles (up to r = 25 $\mu$m)?*

The counting statistics is gets low very quickly with increasing size and thus Poisson uncertainty gets very large (note the already large error-bars in Fig. 9 for the 'not-so-coarse' particles). As we investigate the height dependence, we have to use data from ascends and descends of the Falcon where the sampling time per height range is

limited as described in the text.

*14/1-3: Again, where is this effect of size on particle settling based on? What determines these boundaries?*

We added reference to Eq. 3 and Fig. 1. These boundaries are selected from those provided by the manufacturer.

*14/9: This seems like a very low amount of particles per size bin, how reproducible are these numbers? Have more of similar flights been performed in this area to compare the data to?*

The limited reproducibility, as a result of the low number concentration of coarse particles, is related to the error bars shown in Fig. 9. These error bars show the range where the "real" ratio lies with 95 % probability.

We added two additional profiles from SALTRACE (22 June and 10 July). Our conclusions from the profiles with respect to H1 are the same as for 11 July.

*14/22-23: If this data from SALTRACE is available, why not look at more case studies as in section 4? Since every case is unique, this could give more general insights in the processes involved during transportation of dust within the SAL, related to convective mixing. Why was the case of July 11th chosen for comparison with modeled data?*

We had included a second case study in an earlier version of the manuscript. However we preferred to remove it because not much new insights could be gained from it. July 11th was chosen because of the favorable measurement conditions and a strong dust outbrake arriving over Barbados.

*15/5-6: 'We again analyze the upper 1 km of the SAL [. . .]', I assume this was also done for averaging the POLIS data, however it is not mentioned in the previous section (5.1).*

The reviewer's assumption is correct.

*16/9-10: 'No significant differences in the average profiles were found between day-time and night-time data (not shown).' What does this tell us? That there seems to be convection during both night and day? (which is not assumed in your model)*

Within the uncertainties of the measurements there seems to be no difference between day and night. This could mean that there is also vertical mixing during the night and settling during the day. However, given the uncertainties of the data we can not conclude on that.

*17/19-23: To me it seems that the agreement between the POLIS data and H2 is also not good, just as the bad fit with H1. With the CALIOP data, however, the fit with H2 is much better.*

Also the in-situ data can be much better explained with H2. The reason why we think H2 fits better with the averaged POLIS data was that the height dependence in the upper part (upper 100-200 m) of the SAL better agrees with H2 than with H1. We removed the average POLIS profile from the manuscript because the selection of the cases with a distinct SAL and the selection of its top height was done manually (and developing an objective approach for this procedure is a tricky task).

*17/23-24: You seem to discuss the comparison between CALIOP and H2 also in the next paragraph, so maybe this sentence is a bit redundant here.*

As this is a result we like to keep it here and to refer to the discussion in the next paragraph.

*18/10-14: Could you say a bit more about the supplementary data? If not here, then in the supplement itself?*

We extended the description in the supplement.

*18/24-25: As mentioned for the previous sections as well, I think the results when comparing the model to POLIS data are not that conclusive, for both the case study and the average POLIS data.*

The in-situ data looks more conclusive and the in-situ data was accounted for in this sentence. We are more specific now.

*18/28-29: This relation with particle size isn't really mentioned in the discussion before (except for Figure 9).*

We removed this sentence.

*Technical corrections:*

*9/8: do you mean '[. . .] the importance of the shape dependence ON the gravitational settling for $\delta_l$ profiles.'?*

We reformulated the sentence.

*11/15: 'The relative humidity AT 4500 - 4600 m [. . .]'*

Corrected.

*16/Fig 10 (caption line 4): The statistical uncertainty of the mean, do you mean standard deviation? Also, in the text you refer to CALIOP data, however in the legend now it says CALIPSO, which is not very consistent.*

It is the uncertainty of the mean value. The uncertainty of individual profiles is much larger, mainly because of low signal-to-noise ratios. We use the uncertainty provided by NASA for the individual profiles to estimate the uncertainty of the mean value. We now provide the formula used for the calculation of the uncertainty of the mean.

We corrected the legend.

**Response to reviewer 2**

*The authors investigate processes affecting long-range particle transport in the Saharan Air Layer (SAL) by modeling aerosol property profiles for two scenarios: (1) gravitational settling only and (2) gravitational settling alternating with convective mixing in the SAL. Model results are compared with ground- and space-based lidar measurements as well as aircraft particle counter measurements to determine the most likely scenario.*

*The paper is overall very well written and a valuable contribution that is suitable for publication in ACP. Nevertheless, I have some comments (below).*

We thank also reviewer 2 for his/her thoughtful comments that helped us to improve our manuscript.

*(1) The authors initialize their model with a reference ensemble. As a result, there is no variation of the modeled profiles for different time periods (see for example Figures S-1 to S-5). Is there any way to initialize the model with measured profiles (for example using CALIPSO data) and then compute the profile changes based on the assumptions outlined in the model description?*

In principle, the initial aerosol ensemble in our model could be modified as function of time (and source) if sufficient data about the aerosol near the source is available. In our case such data was not available thus we had to rely on data measured near the desert during SAMUM and before (as incorporated in OPAC) providing some 'typical desert aerosol'. As we need the microphysical properties of the initial aerosol ensemble, we can not use CALIOP data for this purpose. The CALIOP data doesn't provide enough information about the aerosols to invert their microphysical properties.

*(2) The description of the computation of the fraction of removed particles (p. 5, l. 4-14) is not clear to me. I am not sure how Equation 5 has been derived and why $H\_scale$ has been set to 10 km. Some further detail might be beneficial. Also, it is said that $f(r)$ is calculated for $z\_fallen(r) < H\_SAL$ with $H\_SAL$ being the SAL depth. Have the*

*authors considered a starting height of the particles (not all start from the top)? In the context of l. 4 on p. 5, it might also be worth stating that only one air parcel (vertically reaching throughout the SAL) is considered and horizontal mixing is being ignored.*

In our model not all particles start from the top. At start the particles are distributed over the complete (well-mixed) layer. We improved the description and hope it is clearer now. Furthermore, we found an error in the description of Eq. 5 (it calculates the fraction remaining in the SAL, not the fraction removed)

*(3) Two hypotheses are considered: (H1) profile changes are caused by gravitational settling only; and (H2) profile changes are caused by gravitational settling during the night and convective mixing during the day. Do I understand correctly, that gravitational settling is being neglected during the day in H2? I would think that gravitational settling occurs always, irrespective of whether or not convective mixing occurs in addition. Looking at Fig. 10, I could imagine that the additional consideration of daytime gravitational settling in H2 might reduce delta_l at large heights (small dz) and thus lead to a "smoother" profile which compares better to the measured profiles.*

Yes, gravitational settling is neglected during the day in H2. We agree that consideration of settling and imperfect mixing during the day should improve the agreement in Fig. 10. To consider this, we would have to add some parameterization of the strength of the vertical mixing as function of dz. However, we prefer not to further extend our model in the current work, for which the main conclusion is that assuming no vertical mixing of air (H1) is unrealistic. But in future work it seems natural that our, still very idealized, H2 could be replaced by a more realistic model for the SAL.

*(4) The cross-sectional particle radius, r_c, is being used for the computation of the drag force (Section 2.1). However, r_c varies with particle orientation. Has this been accounted for in the model? As the authors assume a random particle orientation, the drag force might have to be calculated as an average over the drag force of single particles with different orientation. Is there any information about whether or not particles*

*are being oriented randomly in nature or if the perhaps align with the flow in some way?*

The reviewer is right in that calculating the drag force for average $r_c$ could differ from the average over the drag force of a single particle in different (random) orientation. However, because of $F_d$ being proportional to $r_c$ (Eq. 2) the difference between both approaches should be rather small.

Small particles usually have random orientation due to Brownian motion. If non-spherical particles are larger, they have the tendency to orient horizontally while sedimenting (i.e. with their larger cross section against the flow). Now after the reviewer's comment we searched the available literature to estimate at which size the particle orientation might become non-random. We found the paper of Ulanowski et al. (2007, doi: 10.5194/acp-7-6161-2007) to provide the required formula. In Fig. 9 of their paper they show that (in the absence of an electric field, 0 V/m) the orientation becomes non-random for prolate spheroids with aspect ratio 1.5 if the maximum particle dimension reaches 10 micrometers. Using their Eq. 14 and 16 and our Eq. 3 we calculated the mean angle for prolate spheroids with different aspect ratios, as function of $r_c$ and plotted it in Fig. 1 at the end of this document.

It shows that the mean angle does not depend very much on the aspect ratio. Assuming that real dust particles (with average aspect ratio 1.6-1.8) behave similar to prolate spheroids we estimate that the transition region between random and non-random orientation is at about $r_c = 5\ \mu$m.

Preferred horizontal orientation would lead in tendency to a reduction of the settling velocity because of increased drag compared to the drag for random orientation. In the book 'Bubbles, drops, and particles' by Clift et al. formulas are provided allowing us to calculate the ratio between the drag force of spheroids in horizontal orientation and the drag force in random orientation. In case of oblate spheroids this ratio is about 1.09 and 1.15 for aspect ratios 1/2 and 1/3, respectively. For prolate spheroids we find ratios of 1.06 and 1.12 for aspect ratios 2 and 3, respectively. Unfortunately, we have

no further information for the case of irregular particle shapes. Given the available information (including the information that average dust aspect ratios are around 1.6-1.8) we estimate that in reality the average settling velocity of desert dust particles at r $> 5$ $\mu$m might be, on average, 5 % smaller in case of horizontal orientation (compared to random orientation). As this reduction does hardly change the modeled lidar profiles we only briefly mention this effect in the paper but do not consider it in the calculations.

*(5) The authors suggest that convective mixing in the SAL occurs due to 'absorption of sunlight by the aerosol particles'. Would this not lead to stronger heating at the top of the SAL compared to the bottom and thus to a stabilization?*

Yes, as the sunlight becomes weaker towards the bottom of the layer a stabilization should be expected. On the other side, if particle settling had started, the amount of absorbers gets lower near the top, which reduces the heating there. Thus, sunlight could lead to a destabilization of the upper part of the SAL under certain conditions.

The main purpose of H2 is to contrast the scenario with no vertical mixing (H1) showing that some kind of vertical mixing occurs in the SAL. H2 probably is not fully realistic and mechanisms other than absorption of sunlight could be relevant for the vertical mixing in nature, as discussed towards the end of the paper. We modified the manuscript (including the title) to make the purpose of the paper more clear.

*(6) Oceanic measurements (e.g. van der Does et al. 2016 [http://www.atmos-chem-phys-discuss.net/acp-2016-344/]) suggest, that particles of a few tens of microns can still be transported some distance across the Atlantic. Do the authors have any evidence of this from their measurements? If that is the case, then the Stokes regime might not apply anymore (for the largest particles). How would this affect the modeled profiles?*

As the number concentration of such large particles is very low, lidar measurements are not sensitive to them and airborne in-situ measurements need a long sampling time to get good statistics about their abundance. However, the airborne in-situ measurements

collected during SALTRACE near Barbados suggest that a few particles with diameters around 30 micrometers arrive in the SAL at that location. For the possibility of transport of 'giant particle' over such long distances we like to refer to our response to reviewer 1 with respect to the error of the Stokes law, to the orientation effects discussed above, to Fig. 3 of our paper, and to the fact that 'extreme shapes' (with small $\xi_{vc}$) could 'survive' longer than those considered in our paper. As the sensitivity of lidar to 'giant particles' is low, the discussed uncertainties about these 'giant particles' hardly affects the modeled profiles.

*Minor comments:*

*(1) P. 2, l. 9-10; To me, the statement 'cannot explain their measurements with Stokes gravitational settling alone' suggests that Stokes settling might be too weak. However, the subsequent statement, that they had to 'reduce the Stokes settling velocity' suggests otherwise. Perhaps rephrasing would clarify this.*

We clarified this sentence.

*(2) P. 3, l. 27; Suggest using half-blanks between unit-parts to avoid Pa (times) s being read as Pas.*

Improved.

*(3) P. 4, l.1; Setting F_g = F_d suggests that a particle would be suspended and not that it would be in 'still air', would it? Unless still air is interpreted such that the particle experiences no vertical movement.*

We made this sentence more clear by writing "... results in a settling velocity of the particle relative to the ambient air of ...".

*(4) P. 6, l. 10; Suggest using micrometers instead of nanometers for consistency.*

Changed.

*(5) P. 7, l. 27; The sentence 'We find a decrease of linear depolarization ratio delta_l*

*with height' is somewhat confusing as it is not clear if the authors mean with decreasing or increasing height (or dz). Perhaps reword. This occurs also at other places in the paper (e.g. p. 9, l.4)*

We changed throughout the paper as suggested by the reviewer.

*(6) P. 9, l. 2; Please clarify which 'smaller-scale features' are meant.*

With 'smaller-scale feature' we refer to the local maximum of $\delta_l$ for shape A at dz=100m. However, as this is not an important aspect of the manuscript, we removed this sentence.

*(7) P. 9, l.9; Why do two particle shapes need to be mixed within one ensemble? Could the shape-dependence of settling not be investigated using two model ensembles each having only particles of a particular shape? (Perhaps I misunderstood the sentence).*

No, this is not possible as we want to investigate the importance of the shape-dependence of settling velocity, which could lead to a shape separation of an initially well-mixed aerosol as suggested by Yang et al. (2013, doi: 10.1002/grl.50603). We try to explain this better now.

*(8) P. 12, l.1; Suggest moving '(thin red line)' to after 4740 m.*

Changed.

*(9) P. 17, l.25; is instead of in*

Corrected.
* * *
[Figure]

Fig. 1.

---

## Referee Report (RR1)

In general, the paper was improved and the writing more clear. However, the paper is still very technical, and for someone not in the field of modeling, like myself, it is quite hard to comprehend the different aspects and various details of the paper. I still have some remaining comments about the paper, listed below.

The authors respond to the question about giant particles that this is not relevant for the present study, but I agree with Reviewer 2 and also would like to refer to the paper by van der Does et al. (2016) for the relevance of so-called "giant particles". It would be very interesting to show proof of why these "giant particles" can be transported up to these great distances, by the mechanism of vertical mixing which the authors propose in the manuscript.

The SALTRACE paper by Weinzierl et al. is mentioned, however it should say "(submitted)" instead of "(2016)", since it is not published, and therefore: not accessible yet. Examples found at: 2/29-30, 3/14, 13/4.

A different example is Haarig et al. ("2017"): it is perhaps somewhat unconventional to cite unpublished work this way, especially unsubmitted work; 15/11.

15/26-27: For the SALTRACE data, the authors now added maps of the position of the aircraft during the sampling period, however it would also be interesting to see elevation profiles of these flights as indeed the altitude at which particles are being transported is very important.

Technical comments:

Several times "Western Atlantic" is written, which should be "western Atlantic". Found at: 1/6, 1/7, 1/19, 3/4, 19/26 (same with "central Atlantic")

2/11: "Particle size distribution" should be "particle-size distribution". Also found at 2/19.
2/19: "[…] in the Caribbean indicates […]" (+ s)
8/12 and 8/28: Here the term "convective mixing" is still used, however in the rest of the paper this is now called "vertical mixing".
11/5: The authors state here "as assumed in literature", could you give examples of such literature?